



# Dynamics of water fluxes and storages in an Alpine karst catchment under current and potential future climate conditions

Zhao Chen[1], Andreas Hartmann[23], Thorsten Wagener[3], Nico Goldscheider[1]

[1]Institute of Applied Geosciences, Karlsruhe Institute of Technology (KIT), Karlsruhe, Germany.
[2]Institute of Hydrology, Albert-Ludwigs-University of Freiburg, Freiburg, Germany.
[3]Department of Civil Engineering, University of Bristol, UK

Corresponding author: Zhao Chen (zhao.chen@kit.edu)

**Abstract.** Climate change projections indicate significant changes to precipitation and temperature regimes in European karst regions. Alpine karst systems can be especially vulnerable under changing hydro-meteorological conditions since
snowmelt in mountainous environments is an important controlling process for aquifer recharge, and is highly sensitive to varying climatic conditions. The current study presents an investigation of present and future water fluxes and storages at an Alpine karst catchment using a distributed numerical model. A delta approach combined with random sampling was used to assess the potential impacts of climate changes. The study site is characterized by high permeability (karstified) limestone formations and low permeability (non-karst) sedimentary flysch. The model simulation under current conditions
demonstrates that a large proportion of precipitation infiltrates into the karst aquifer as autogenic recharge. Surface runoff in the adjacent non-karst areas partly infiltrates into the karst aquifer as allogenic point recharge. Moreover, the result shows that surface snow storage is dominant from November to April, while subsurface water storage in the karst aquifer dominates from May to October. The climate scenario runs demonstrate that varied climate conditions significantly affect the spatiotemporal distribution of water fluxes and storages: (1) the total catchment discharge decreases under all evaluated
future climate conditions. (2) The spatiotemporal discharge pattern is strongly controlled by temperature variations, which can shift the seasonal snowmelt pattern, with snow storage in the cold season (December to April) decreasing significantly under all change scenarios. (3) Increased karst aquifer recharge in winter and spring, and decreased recharge in summer and autumn, partly offset each other. (4) Impacts on the karst springs are distinct; the permanent spring presents a "robust" discharge behavior, while the estavelle is highly sensitive to changing climate. This analysis effectively demonstrates that the
impacts on subsurface flow dynamics are regulated by the characteristic dual flow and spatially heterogeneous distributed drainage structure of the karst aquifer. Overall, our study suggests that bespoke hydrological models tailored to the specific subsurface characteristics of an Alpine karst catchment are needed to understand climate change impact.

## 1. Introduction

The Alps, called the "water tower of Europe", form headwaters for important regional river systems (Viviroli et al., 2007).
Alpine catchments are generally characterized by above-average precipitation due to orographic effects, as well as by colder



temperatures resulting in lower evapotranspiration and temporary water storage in the form of snow and ice (Zierl and Bugmann, 2005). Climate projections indicate that a shift in snow and precipitation patterns is likely to alter catchment runoff regimes (Gobiet et al., 2014). Additionally, extreme events, such as floods and droughts, are expected to increase in frequency and intensity (Dobler et al., 2013; Rössler et al., 2012). For sustainable management of water resources in Alpine

areas, it is imperative to understand the complex mountain hydrological processes (Kraller et al., 2012).

In this context, numerical models are usually applied to describe the hydrological processes in Alpine catchments (Abbaspour et al., 2007; Achleitner et al., 2009; Benischke et al., 2010; Braun and Renner, 1992; Junghans et al., 2011; Kraller et al., 2012). Lumped conceptual simulation models are easy to use in gauged catchments because their parameters can be effectively found via calibration. On the other hand, distributed simulation models are required for studying the

spatial patterns of hydrological processes across a catchment. However, spatially-distributed models face challenges in Alpine areas concerning the assessment of input variables and model parameters (Kraller et al., 2012; Kunstmann and Stadler, 2005). Furthermore, most distributed models focus on surface hydrological variables (e.g. vegetation, soil and snow cover) or/and anthropogenic variables (e.g. land use and water use), with relatively poor subsurface representations. Few studies (e.g. Kraller et al., 2012; Kunstmann et al., 2006; Kunstmann and Stadler, 2005) explicitly considered subsurface

processes such as recharge, drainage and storage in their models. It is generally accepted that the geological and lithological setting for mountainous catchments are often complex and could have significant impact on the catchment flow regime (Goldscheider, 2011; Rogger et al., 2013). The situation is even more complex when mountain ranges within a catchment consist of highly permeable limestone formations characterized hydraulically by fissures and/or conduit drainage networks, and concentrated discharge via springs (Goldscheider, 2005; Gremaud et al., 2009; Lauber and Goldscheider, 2014). In order

to better understand complex hydrological processes at mountainous karstic catchment as well as quantify their dynamics, this study presents a spatially-distributed investigation of the water fluxes and storages in a high-elevation Alpine catchment considering its complex subsurface heterogeneous drainage structure. The study catchment constitutes an optimal test case to explore complex hydrological processes since it includes many typical characteristics of Alpine catchments, such as a seasonal snow cover, a large range of elevations and a highly varied catchment flow regime. Furthermore, the hydrogeology

in the investigated catchment is complex. It is characterized by high permeability limestone formations (karst areas) and low permeability flysch[1] sedimentary rocks (non-karst areas) as described by Goldscheider (2005). Here, we expanded an existing model (Chen and Goldscheider, 2014) by adding a snow accumulation/melting routine with high spatiotemporal resolution. We also developed a tailored calibration strategy, building on a previous sensitivity analysis by Chen et al. (2017), to calibrate the proposed catchment model reasonably and effectively.

Several recent studies indicated the significant impact of climate change on the catchment discharge behavior of Alpine areas, and demonstrated the changing characteristics of flow regimes including amount, seasonality, minima and maxima, as well as impacts on other hydrological variables, e.g. soil moisture and snow cover (Dobler et al., 2012; Jasper et al., 2004;

_____________________

[1] The flysch formations consist of an interstratification of claystone, impure sandstone, marl and thin-bedded limestone



Kunstmann et al., 2004; Middelkoop et al., 2001; Rössler et al., 2012; Zierl and Bugmann, 2005). However, the relationship between subsurface hydrological processes (recharge, storage and discharge) and changing climate conditions has not yet been considered in any detail. Gremaud et al 2009 and Gremaud and Goldscheider 2010 studied a geologically complex, glacierised karst catchment in Alps by combining tracer tests and hydrological monitoring and found that the changing hydro-meteorological conditions affect the water storage in snow and ice significantly, which have high impact on the aquifer recharge processes and discharge dynamics. Finger et al 2013 investigated glacier meltwater runoff in a high Alpine karst catchment under present and future climate conditions using tracer experiments, karst structure modeling and glacier melt modeling. The results indicated that parts of the glacier meltwater are drained seasonally by underlying karst system and the expected climate change may jeopardize the water availability in the karst aquifer. In order to better understand climate change effects on complex hydrological processes in Alpine karstic environment, we assessed the impacts of varied climate conditions on the water fluxes and storages in the simulated model domain, and we identified the hydrological processes most sensitive to potential climate change. For this analysis, we used a pragmatic and widely used delta approach to project the climate change in the model domain (e.g. Dobler et al., 2012; Lenderink et al., 2007; Singh et al., 2014).

## 2. Study area

The study catchment is located in the northern Alps on the Germany/Austria border (Fig. 1a). It has an area of about 35 km$^2$, and an altitude varying between 1000 m asl (the lowest part of the Schwarzwasser valley) and 2230 m asl (the summit of Mt. Hochifen). The climate in the area is cool-temperate and humid. The nearest permanent weather station lies to the east in the Breitach valley at an altitude of 1140 m asl. There, the mean monthly temperature ranges from -2.2 °C in January to 14.4 °C in July, with an annual average of 5.7 °C (based on data from 1961 to 1990, available from Water Authority Vorarlberg). The mean annual precipitation is 1836 mm with a maximum in June-August and a secondary maximum in December-January. Snow accumulates commonly between November and May.

Hydrogeologically, the investigated catchment can be divided into karst and non-karst areas, whose boundary is more or less marked by the Schwarzwasser river. The karst area is characterized by the highly permeable Schrattenkalk limestone formation (with about 100 m thickness), which is underlain by marl formations. The underground flow paths in the karst system are controlled by local folds and follow plunging synclines. The karst aquifer system discharges in several springs (a permanent spring QS, a large but intermittent overflow spring QA and an estavelle[2] QE) at different elevations (and recharged directly from precipitation) as well as indirectly in surface streams that drain the non-karst area. These are formed by low to moderately permeable flysch sedimentary rocks. Several quantitative multi-tracer tests (Goldscheider, 2005; Göppert and Goldscheider, 2008; Sinreich et al., 2002) revealed two parallel drainage systems in this valley: a surface stream

---

[2] Opening in karstic terrane which acts as a discharge spring during high flow conditions and as a swallow hole during low flow conditions



and a continuous underground karst drainage system along the valley axis, which are hydraulically connected in the upper part of the valley.

## 3. Methodology

### 3.1 Setup of the catchment model

Our model is based on the existing distributed karst catchment model by Chen and Goldscheider (2014), which in turn has been derived from the distributed hydrologic-hydraulic water quality simulation model – Storm Water Management Model (SWMM, version 5.0) developed by the U.S. Environmental Protection Agency (Rossman, 2010). The hydrological conceptual model was developed mainly based on the geologic study by Wagner (1950), the speleological investigation by the regional caving club (Höhlenverein Sonthofen, 2006) and numerous hydrogeological field experiments by Goldscheider

(2005). Additional tracer experiments by Göppert and Goldscheider (2008) and Sinreich et al. (2002) improved this conceptual model.

Compared to the existing karst catchment model, new developments are: (1) the model domain is extended to the non-karst area of our study site to consider the surface runoff generated from low permeability flysch formations, which can infiltrate into the underground karst drainage network in the upper part of the valley (Fig. 1c and 2a), (2) we considered the slow flow

for individual karst sub-catchments, which should approximately represent long term matrix flow (Fig. 2), and (3) the space- and time-varying snow accumulation and melt are included (described in section 3.3). In line with these changes, the whole model domain is divided into 4 karst sub-catchments due to underground drainage systems and 2 non-karst sub-catchments due to surface streams, which consist of 29 sub-units, divided by 6 elevation bands. The karst and non-karst catchments are hydraulically connected, i.e. the underground karst drainage conduits are connected with the surface stream channels in the

upper part of the valley. In total, 76 model parameters (Supplementary material) are considered for the model setup: (1) Model parameters x1 – x20 define the main hydrological processes of the unsaturated zone in the individual karst sub-catchments and the top layer of the low permeable flysch rocks, (2) model parameters x21 – x76 describe the geometry and hydraulic properties of the karst drainage conduit network as well as surface stream channels in the non-karst area.

### 3.2 Monitoring network and data availability

Four observation locations in the studied catchment were considered here: (1) QS at 1035 m asl in the valley, (2) QA at 1080 m asl, (3) QE at 1120 m asl and (4) a gauging station (SR) at 1122 m asl quantifying the surface runoff from the upper part of Schwarzwasser valley. Hourly measured discharges at the above-mentioned monitoring stations are used, whereas the measurements for QS and QA are available from November 2013 to October 2014, for QE and SR only from July to October 2014. For the same period, we interpolated the meteorological data (hourly precipitation, air temperature and relative

humidity) from nine weather stations (Fig. 1b) across the study catchment at a 100 m × 100 m grid resolution using combined inverse distance weighting and linear regression gridding. Mean areal precipitation and potential





evapotranspiration for individual sub-units are determined based on the interpolated meteorological data, in which hourly potential evapotranspiration is estimated using a modified Turc-Ivanov approach after Wendling and Müller (1984).

## 3.3 Modeling snow accumulation and melting

We adopted a simple, widely used (e.g. Bergström, 1975; Kollat et al., 2012; Seibert, 2000) degree-day approach to modeling snow. We further modified the calculation of snowmelt using the approach proposed by Hock (1999), to simulate more realistic hourly varied snow melting in mountainous catchments:

$$M = \begin{cases} (MF + \alpha \times I) \times (t - Ts), & t > Ts \\ 0, & t \leq Ts \end{cases} \tag{1}$$

Where M is snowmelt (mm h$^{-1}$), MF is melt factor (mm h$^{-1}$ °C$^{-1}$), $\alpha$ is radiation coefficient, I is potential clear-sky direct solar radiation at surface (W m$^{-2}$), t is measured hourly air temperature (°C) and Ts is threshold temperature (°C) for snow melting. The melt factor and the radiation coefficient are empirical coefficients and can be estimated by model calibration. The distributed potential clear-sky direct solar radiation is dependent on surface topography and calculated with 100m × 100m grid resolution for the investigated area using the approach developed by Kumar et al. (1997).

## 3.4 Model calibration

### 3.4.1 Model optimization

We used the DiffeRential Evolution Adaptive Metropolis (DREAM) by Vrugt (2016) to calibrate the model. The simultaneous minimization of the sum of the squared errors (SSE) of multiple observed time series was applied to constrain the model parameter space (described in section 3.4.2), which was defined based on our previous experience in the study region (Chen and Goldscheider, 2014; Chen et al., 2017). The DREAM algorithm allows an initial population of parameter sets to converge to a stationary sample.

### 3.4.2 Calibration strategy

In a previous comprehensive sensitivity analysis we demonstrated that the controlling parameters exhibit varying sensitivity for different hydrodynamic conditions and for different spatially-distributed model outlets (Chen et al., 2017). Based on this information, we designed four steps to calibrate the model using different hydrodynamic system conditions and the observed time series for different outlets. Additionally, to explicitly consider or completely remove the snow dynamic during calibration, we divided the whole simulation period into a snow period (November 2013 – June 2014) and a rainfall period (June 2014 – October 2014). There was no snow cover anywhere in the catchment during the rainfall period.

The multi-step calibration procedure applied here is illustrated in Figure 3. In step 1, we used the rainfall period to constrain the model parameters of the unsaturated zone and the drainage network during medium and high flows. The different hydrodynamic conditions are defined using the exceedance probability of the observed discharge at QS. In step 2, we used the snow period to constrain the parameters of snow storage during medium and high flows, whereas in the observation data





the snow accumulation and melting dynamics in the catchment are clearly reflected. The time series of QS and QA are used for this calibration step. In step 3, we focused on the low flows in the same simulation period as during step 2 to further constrain the parameters of storage in snow, unsaturated zone and drainage network using the observation data of QS and QA. In step 4, the ranges of the previous parameters were constrained continuously using all flow conditions and observation time series from all four outlets.

The error function used in DREAM is the sum of the SSE values defined in individual calibration steps (Eq. 3 for step 1 and 4; Eq. 4 for step 2 and 3):

$$SSE = \sum_{t=1}^{N}(Q_{t,o} - Q_{t,s})^2 \tag{2}$$

Where $Q_{t,o}$ is the observed discharge at time step t, $Q_{t,s}$ is the simulated discharge at time step t and N is the number of measurements in the selected time series.

$$SSE_{Objective1} = SSE_{QS} + SSE_{QA} + SSE_{QE} + SSE_{SR} \tag{3}$$

$$SSE_{Objective2} = SSE_{QS} + SSE_{QA} \tag{4}$$

For each calibration step, 5000 parameter sets were generated using Latin Hypercube sampling within the defined prior parameter ranges. The last 1000 parameter sets of the converged sample in each calibration step are used to represent the posterior distribution of "best" parameter sets. Posterior parameter bounds are determined using the 95 % confidence interval for these 1000 parameter sets. The parameter bounds of a previous step were adopted as a-priori parameter bounds for the subsequent calibration step.

**3.5 Estimation of water storage**

To understand water storage processes within the catchment, we estimated the temporary water storage volumes for the entire catchment (Eq. 5), karst area (Eq. 6) and non-karst area (Eq. 7):

$$S_{t,catchment} = \sum_{t_0}^{t}(P_{t,catchment} - ET_{t,catchment} - Q_{t,catchment}) \tag{5}$$

$$S_{t,karst} = \sum_{t_0}^{t}(P_{t,karst} + R_{t,allogenic} - ET_{t,karst} - Q_{t,karst}) \tag{6}$$

$$S_{t,nonkarst} = \sum_{t_0}^{t}(P_{t,nonkarst} - R_{t,allogenic} - ET_{t,nonkarst} - Q_{t,nonkarst}) \tag{7}$$

Surface runoff from the non-karst area can infiltrate into the underground karst drainage network because the non-karst and karst areas are hydraulically connected in the upper part of the valley. Infiltration is considered as allogenic recharge for the karst area and was taken into account for the storage calculation for the non-karst area. Additionally we simulated the temporary subsurface water storage volume for the karst aquifer (Eq. 8):

$$S_{t,karstaquifer} = \sum_{t_0}^{t}(R_{t,autogenic} + R_{t,allogenic} - Q_{t,karst}) \tag{8}$$

Where $S_t$, $P_t$, $ET_t$, $R_t$ and $Q_t$ are the storage, precipitation, evapotranspiration, recharge and discharge in volume at time step t ($t_0$ is first simulation time step). The simulated temporary storage volumes for the whole catchment ($S_{t,catchment}$), karst area ($S_{t,karst}$) and karst aquifer ($S_{t,karstaquifer}$) are not the absolute volumes, as the calculation is referred to the initial water storage volume in the karst aquifer, which is set at $t_0$ and cannot be taken into account.



### 3.6. Climate change projections

The focus of this analysis is to quantify the impact of varying climate conditions on the water fluxes and storages throughout the model domain and to identify the hydrological processes most sensitive to potential climate change within the study catchment. We chose the probabilistic scenarios of precipitation and temperature by Frei (2004) for the northern Alps as the basis for our study. The median values (q0.5) and the confidence intervals (q0.025 to q0.975) of the probabilistic scenarios for years 2030, 2050 and 2070 were derived in Frei (2004) and given in Table 1. We used a delta approach to project the potential climate change scenarios in the investigated catchment by changing precipitation and temperature time series for the pre-defined months (December-February, March-May, June-August and September-November) by a given delta (percentage or value). For the analysis, we first focused on the median climate scenarios of 2030, 2050 and 2070 (described in section 4.3.1) to better understand the general trend of the climate change projections. In the second part of the analysis, we considered the uncertainty in the climate scenario for 2070 and estimated its impact on the simulated water fluxes and storages across the model domain (described in section 4.3.2). To consider the climate change scenario uncertainty, 1000 uniformly distributed random samples within the defined confidence intervals for the deltas of precipitation and temperature are used.

### 4. Results

### 4.1 Model performance

Figure 4 shows the simulated karst spring discharges as well as the surface runoff generated from the non-karst area of the final calibrated model. The transient and highly variable discharge behavior at the four spatially-distributed model outlets is simultaneously simulated at an hourly time step. The quality of the model simulation is demonstrated by two different statistical criteria, RMSE and Nash–Sutcliffe Coefficient (NSC): RMSE values are 0.118 m3/s for QS, 0.448 m3/s for QA, 0.419 m3/s for QE and 0.248 m3/s for SR. NSC values are 0.71 for QS, 0.80 for QA, 0.74 for QE and 0.66 for SR.

### 4.2. Estimated water fluxes and storages

For a simulation period of about 330 days, we estimated that about 5 % of the total precipitation (52.79 MCM[3]) left the catchment as evapotranspiration (2.39 MCM) (Fig. 5). Furthermore we calculated that about 84 % of the recharge (44.02 MCM) to the karst aquifer is contributed by diffuse infiltration (36.78 MCM) over the karst area. The remaining 16 % of the recharge is contributed by the allogenic recharge (7.24 MCM); i.e. direct infiltration of the surface runoff from the non-karst area into the underground karst drainage network in the upper part of the valley. The catchment is mainly drained by the karst springs. About 20 % of the total catchment discharge (49.41 MCM) is provided by QS (10.09 MCM), 44 % by QA (21.81 MCM), 23 % by QE (11.29 MCM) and 13 % by the surface runoff (6.23 MCM).

---

[3] MCM for million cubic meters





We compared the estimated water storages for the whole catchment, karst area, non-karst area and karst aquifer to better understand different storage processes (snow storage, soil water storage and subsurface water storage) in the model domain (Fig. 6). It is considered that in the simulated winter and early spring (November 2013 – March 2014), the catchment water storage is mainly characterized by snow storage in both the karst and non-karst areas. Afterwards, snow melt (April – May 2014) led to rapidly decreasing catchment snow storage, but increasing storage in the karst aquifer as subsurface water in both fast and slow paths. During the rainfall season in the simulated summer and autumn (June – October 2014), the catchment storage is mainly characterized by subsurface water storage in the karst aquifer, while during medium and high flows the water is also stored intermittently in the top layer of the non-karst area.

## 4.3. Assessing the impact of climate projections

An overview about the change in water fluxes and storages under changing climate conditions (median climate scenarios and uncertainty of the climate scenario 2070) is given in Table 2.

### 4.3.1 Median climate scenarios

The simulations (Fig. 7-9) show that the water fluxes and storages are sensitive to varying climate conditions. Compared to the current situation, the precipitation over the catchment area is gradually decreasing (medians of -4.2 %, -8.2 % and -11.0 %) for the climate scenarios of 2030, 2050 and 2070, whereas the evapotranspiration is increasing (medians of +5.5 %, +11.4 % and +16.0 %). The modeled precipitation, temperature and evapotranspiration for future contribute to the decreased recharge (medians of -4.4 %, -8.8 % and -12.0 %) to the karst aquifer, whereas the recharge pattern is shifted, i.e. the recharge is increasing in winter and spring and decreasing in summer and autumn (Fig. 7).

Furthermore, the catchment water storage pattern changes significantly, especially during the normally "cold" period (from January to April). Under the current condition, maximal 6.50 MCM water is stored in snow, whereas at the same time, only 3.27 MCM as snow storage is estimated there under the conditions of 2070 (Fig. 8). This indicates that the simulated future climate conditions affect the snow storage massively. Comparatively, the catchment water storage during the rainfall season is much less influenced. For the karst aquifer, the shift of recharge pattern towards increased recharge in winter and spring, and decreased recharge in summer and autumn produces compensation, i.e., the annualized balance between recharge and discharge for the karst aquifer is constant for the simulations of 2030, 2050 and 2070. Furthermore, the influence of the varied climate conditions on the intermediate water storage in the karst aquifer (epikarst and fast flow path) and top layer of the non-karst area are limited.

Our simulations (Fig. 9) show that the catchment discharge amount varies under changing climate conditions. The total discharge of QE is decreasing gradually (medians of 9.1 %, -19.0 % and -27.6 %) for 2030, 2050 and 2070, compared to the current situation. However, the deficit for QA (medians of -2.1 %, -3.8 % and -4.2 %) and QS (medians of -2.0 %, -3.9 % and -5.2 %) is less significant. For the total surface runoff generated from the non-karst area, climate change effects are clearly perceptible with the total runoff decreasing (medians of -6.4 %, -11.4 % and -15.1 %) for 2030, 2050 and 2070. Also,




the catchment discharge pattern is influenced significantly. The simulated increasingly warming winters and springs from 2030 to 2070 shift the discharge pattern of QA, QE and surface runoff continuously, while the discharge pattern of QS is quite stable until 2070.

### 4.3.2 Uncertainty of the climate scenario 2070

The results show (Fig. 7) that the impacts of the possible climate scenarios for 2070 on the precipitation, evapotranspiration, recharge and catchment discharge are uncertain. Compared to the current situation, a general trend with the decrease of precipitation, recharge and catchment discharge or with the increase of evapotranspiration can be expected. In the most extreme cases, the change of precipitation varies between -26.4 % and 0.7 %, evapotranspiration between -1.8 % and 39.6 %, recharge to the karst aquifer between -27.1 % and -0.6 % and catchment discharge between -25.5 % and -0.2 %, compared to the current situation. Furthermore, the scenario runs indicate a shift of evapotranspiration, recharge and catchment discharge pattern towards increased recharge as well as catchment discharge in winter and spring and constantly increased evapotranspiration throughout the year.

Moreover, the scenario runs indicate a clear trend with the decrease of water storages for the simulated catchment (Fig. 8). Under the condition "extremely warm" of 2070, the snow storage of the catchment changes so dramatically that almost no water can be stored in snow during the normally "cold" period (from December to April). Simultaneously, the water storage pattern in the karst aquifer can be significantly shifted due to the earlier-starting snow melt. Also, the water storage in the karst aquifer in summer and autumn are influenced strongly due to the significantly decreased recharge. This contributes to a clearly negative "balance" at the last time step of the simulation under the "extremely dry" conditions of 2070. If this negative water storage could be transferred to the coming year, it would cause more negative "balance" for the simulated karst aquifer based on the simulated climate condition. Accordingly, the stored water resource in the karst aquifer would be decreased significantly.

Regarding to the impacts of the uncertain scenarios on the karst spring discharges and surface runoff, distinct trends are identified (Fig. 9): (1) a clear trend with the decrease of QE and SR, (2) impacts on QA are highly uncertain even an increase of its total discharge is projected and (3) impacts on QS are clearly less uncertain and a general trend with decrease of QS can be expected. In the most extreme cases, compared to the current situation, the change of QS varies between -25.5 % and 0.7 %, QA between -18.8 % and 9.9 %, QE between -53.3 % and -10.6 % and surface runoff between -31.3 % and -2.9 %. QS's discharge is considered as the most "robust" in the face of strongly varied climate conditions. Furthermore, a common shift of the discharge pattern of all karst springs and the surface runoff pattern are identified, i.e. increased QS, QA, QE and SR in winter and early spring.



## 5. Discussion

### 5.1. Realism of the model simulations

In this study, the karst catchment model simulates the transient and highly variable discharge behavior simultaneously at the four spatially-distributed model outlets. The evaluation using different statistical metrics indicate that the results are

satisfying. The previous studies proved that the model adequately represents the high permeability flow and flooding mechanisms observed in the studied karst aquifer and is also able to transform them into realistic catchment responses during rainfall periods (Chen and Goldscheider, 2014; Chen et al., 2017). The current study shows that the snow dynamic reflected on the major karst springs (QS and QA) is reproduced in the model. It indicates that the model represents the recharge process driven by the snow accumulation and melting in the studied karst catchment. During the snow accumulation period

(Nov.2013 – Feb.2014), the karst system was under-saturated, and QS discharged the whole catchment, while other karst springs (QA and QE) were dry and no significant surface runoff generated from the non-karst area. The simulation is consistent with our measurements and field observations. It indicates that the model represents the dominant flow process for the investigated karst catchment during low flow conditions. We find that the surface runoff generated from the non-karst area is much less than the effective precipitation for the non-karst area. The reason is that the allogenic recharge leads to

significant loss. This model behavior represents the conceptualization of our understanding about the hydraulic connection between the karst and non-karst areas. However, the model evaluation shows that the model did not very accurately simulate the surface runoff in response to heavy rainfall events. The reason could be that we oversimplified the complex hydrological situation in the non-karst area under-representing its runoff dynamics.

The estimated low evapotranspiration for the investigated catchment seems to be realistic. In the elevated part of the study

area, no significant thickness of soil cover can be considered. Even in the extended karst area, soil cover is missing, the limestone rocks are bare, and the rainfall can directly infiltrate into the karst system through surface features, leading to a high infiltration rate for the karst aquifer (Goldscheider, 2002). However, the total amount of evapotranspiration may be underestimated, as the potential evaporation from snow cover (e.g. Leydecker and Melack, 2000) was not taken into account in our model. Accordingly, the estimated infiltration rate of 95.5 % for the karst aquifer may also be overestimated. For

comparison, Malard et al. (2016) estimated average infiltration rates for mountainous karst catchments across Switzerland varying between 60 % and 90 % of total precipitation using a GIS-based approach.

### 5.2. Identifying hydrological processes sensitive to potential climate change patterns

The climate scenario runs show the water fluxes and storages within the simulated catchment are sensitive to varying climate conditions. Basically, the catchment discharge amount is precipitation driven. The discharge pattern is controlled by the

temporal distribution of precipitation on the one hand, and the temperature pattern on the other hand. The snow storage in the catchment is highly sensitive to the temperature variation, which can shift the seasonal snow melting for the catchment, recharge pattern for the karst aquifer and drainage pattern of the non-karst area. The impacts of potential climate change on




snow accumulation and melting processes have also been reported in other catchments across the European Alps (Horton et al., 2006; Zierl and Bugmann, 2005).

For the karst aquifer, due to its characteristic duality of flow and storage and additional spatially heterogeneous distributed drainage structure, the impacts of the varied climate conditions on QS, QA and QE are distinct. The simulations demonstrate

well that QE is highly sensitive to changing climate conditions. The explanation is that QE acts as the highest overflow outlet of the studied karst aquifer, and its activation is strongly controlled by the hydrodynamic conditions in the karst drainage network, which are in turn highly sensitive to recharge and fast flow processes. In contrast, QS is the lowest outlet for the karst aquifer and its discharge is "guaranteed" by the long term water storage in matrix. Accordingly, QS is the most "robust" in the face of changing climate conditions. Under the simulated climate scenarios, QA shows a mixed character. On

the one hand, QA's discharge is significantly less influenced than QE and on the other hand, QA's discharge pattern can be more easily shifted than QS. It demonstrate well that the high permeability flow in the conduit network with less water storage capacity is sensitive to changing hydrological conditions, while the low permeability flow in the matrix with greater water storage capacity is more resistant. In the non-karst area, the varied climate conditions affect the snow accumulation and melting patterns. As the non-karst and karst areas are hydraulically connected in the upper part of valley, the predicted

earlier-starting snow melt can generate more runoff in the non-karst area which partly infiltrates into the underground drainage network leading to greater loss for the surface runoff and increased allogenic recharge to the karst aquifer.

For the current analysis, we used a pragmatic approach to analyze potential climate change scenarios. The uncertainties of the climate scenarios were considered using a random sampling based approach. The final results indicate the impacts of the seasonal changes in pattern of precipitation and temperature on the spatially varied hydrological processes within the

catchment. Additionally, we investigated the flow exceedance probability of karst springs and surface runoff from the non-karst area (supplementary material) and find that the simulated climate conditions affect the frequency and amplitude of catchment flows. This suggests that the impacts of the temporally stochastic distributions of meteorological parameters and their variability on the catchment flow dynamics should be systematically investigated.

## 6. Conclusion

The current work presents an investigation of the water fluxes and storages in a high-elevation Alpine catchment. We extended the existing karst catchment model developed by Chen & Goldscheider (2014) to consider spatially-distributed snow dynamics and complex surface and subsurface heterogeneous drainage structures. The new model is able to simultaneously simulate the transient and highly variable discharge behavior of four spatially-distributed model outlets at an hourly time step. Furthermore, we estimated the water fluxes and storages within the model domain. The results demonstrate

that the spatiotemporal distribution of water fluxes and storages is controlled by the surface and subsurface hydrological setting. We find a large portion of precipitation infiltrates in the karst aquifer as autogenic recharge and contributes to surface runoff in the adjacent non-karst area, which can partly infiltrate into the karst aquifer as allogenic point recharge. In





the simulation period, the catchment is mainly drained by the karst springs, about 20 % of the total catchment discharge is provided by the permanent spring QS, 44 % by the overflow spring QA, 23 % by the estavelle QE and 13 % by the surface runoff SR generated from the non-karst area. In the simulated winter and early spring (November 2013 – March 2014), the catchment water storage is mainly characterized by the snow storage both in the karst and non-karst areas. During the rainfall season in the simulated summer and autumn (June – October 2014), the catchment storage is mainly characterized by the subsurface water storage in the karst aquifer.

Additionally, we studied the impacts of potential climate change patterns on the spatially varied surface and subsurface hydrological processes in the model using a delta approach combined with a random sampling technique. The scenario runs demonstrate that the varied climate conditions affect the spatiotemporal distribution of water fluxes and storages within the catchment significantly: (1) the total catchment discharge decreases under all evaluated future climate conditions. (2) The catchment snow storage during normally "cold" period from December to April decreases significantly, while the autogenic and allogenic recharge to the karst aquifer increase. (3) In the karst aquifer, due to its storage capacity, the shift of recharge pattern towards increased recharge in winter and spring, and decreased recharge in summer and autumn offset each other under the varied climate conditions. (4) The impacts of the potential future climate conditions on the karst springs are distinct. The permanent spring QS presents a "robust" discharge behavior, while the estavelle QE is highly sensitive to the changing climate conditions. QA's discharge is significantly less influenced than QE and its discharge pattern can be more easily shifted than QS. This demonstrates well that the impacts of potential climate change on the subsurface flow dynamics are regulated by the karst aquifer due to its characteristic dual flow systems and spatially heterogeneous distributed drainage structure.

As our climate scenario projections use a simple delta approach, the impact of temporally stochastic distributions of meteorological parameters and their variability could not be investigated in this study. Accordingly, the results should only be applied to understand the relationship between the hydrological processes within the studied catchment and potential climate change patterns. It would be interesting to use more realistic data, i.e. the precipitation and temperature time series downscaled from regional climate models to investigate their impact on the spatially-distributed water fluxes and storages. But we warn that the measurements of meteorological variables in high-elevation mountainous environment have large quite uncertainty. These uncertainties may have an impact on the model simulations and the understanding of derived processes.

**7. Acknowledgments**

We thank Clemens Mathis and Ralf Grabher from Water Authority Vorarlberg (Austria) for providing data, Laurence Gill (Trinity College Dublin) for inspiring discussion concerning model setup, Joël Arnault (Karlsruhe Institute of Technology) for providing MATLAB routine for the interpolation of meteorological parameters and Timothy Bechtel (Franklin & Marshall College) for proofreading the manuscript.



## 8. Figure captions

Figure 1 a) Location of the study area, b) digital elevation model with grid size 100 m × 100 m for the studied catchment and its surrounding area with weather stations used for the interpolation of meteorological parameters and c) model configuration (modified after Chen and Goldscheider 2014).

Figure 2 a) model concept for the sub-catchments in the non-karst area and b) model concept for the sub-catchments in the karst area.

Figure 3 Strategy for the multi-step model calibration, where LF / MF / HF are for low / medium / high flow conditions.

Figure 4 Observed and simulated discharge of four spatially-distributed model outlets QS, QA, QE and SR using the best calibrated model parameter set for the period November 2013 – October 2014. Additionally, the mean catchment

precipitation and temperature for the same period are shown.

Figure 5 Estimated cumulative volumes of precipitation, evapotranspiration, recharge and discharge for the studied catchment for the period November 2013 – October 2014 on an hourly time step in million cubic meters (MCM).

Figure 6 Estimated temporary water storage volumes for the whole catchment, karst area, non-karst area and karst aquifer for the period November 2013 – October 2014 on an hourly time step in million cubic meters (MCM).

Figure 7 Impacts of the median climate scenarios (q0.5) for 2030, 2050 and 2070 as well as the uncertain climate scenarios (1000 random sampled combinations) for 2070 on the simulated precipitation, evapotranspiration, recharge and discharge for the studied catchment.

Figure 8 Impacts of the median climate scenarios (q0.5) for 2030, 2050 and 2070 as well as the uncertain climate scenarios (1000 random sampled combinations) for 2070 on the simulated water storages of the whole catchment, karst area, non-karst

area and karst aquifer.

Figure 9 Impacts of the median climate scenarios (q0.5) for 2030, 2050 and 2070 as well as the uncertain climate scenarios (1000 random sampled combinations) for 2070 on the simulated discharge of QS, QA, QE and surface runoff from the non-karst area.

## 9. Supplementary material

Figure S1 Impacts of the median climate scenarios (q0.5) for 2030, 2050 and 2070 as well as the uncertain climate scenarios (1000 random sampled combinations) for 2070 on the FDC (0 % – 10 % exceedance probability) of QS, QA, QE and surface runoff from the non-karst area for the time window from December to March.

Figure S2 Impacts of the median climate scenarios (q0.5) for 2030, 2050 and 2070 as well as the uncertain climate scenarios (1000 random sampled combinations) for 2070 on the FDC (80 % – 100 % exceedance probability) of QS, QA, QE and

surface runoff from the non-karst area for the time window from June to October.





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





Table 1a: The median (q0.5) and the confidence intervals (q0.025 and q0.975) of the probabilistic precipitation scenarios for year 2030, 2050 and 2070 are explicitly given as percentage change (compared to 1990) and applied for the analysis described in section 3.6.

| season | precipitation scenario (%) | | | | | | | | |
| --- | --- | --- | --- | --- | --- | --- | --- | --- | --- |
| | 2030 | | | 2050 | | | 2070 | | |
| | q0.025 | q0.5 | q0.975 | q0.025 | q0.5 | q0.975 | q0.025 | q0.5 | q0.975 |
| Dec/Jan/Feb | -1 | +4 | +11 | -1 | +8 | +21 | -1 | +11 | +30 |
| Mar/Apr/May | -6 | 0 | +5 | -11 | -1 | +10 | -15 | -1 | +13 |
| Jun/Jul/Aug | -18 | -9 | -3 | -31 | -17 | -7 | -41 | -23 | -9 |
| Sep/Oct/Nov | -8 | -3 | 0 | -14 | -6 | -1 | -20 | -9 | -1 |

Table 1b: The median (q0.5) and the confidence intervals (q0.025 and q0.975) of the probabilistic temperature scenarios for year 2030, 2050 and 2070 are explicitly given as absolute change (compared to 1990) and applied for the analysis described in section 3.6.

| season | temperature scenario (°C) | | | | | | | | |
| --- | --- | --- | --- | --- | --- | --- | --- | --- | --- |
| | 2030 | | | 2050 | | | 2070 | | |
| | q0.025 | q0.5 | q0.975 | q0.025 | q0.5 | q0.975 | q0.025 | q0.5 | q0.975 |
| Dec/Jan/Feb | +0.4 | +1 | +1.8 | +0.9 | +1.8 | +3.4 | +1.2 | +2.6 | +4.7 |
| Mar/Apr/May | +0.4 | +0.9 | +1.8 | +0.8 | +1.8 | +3.3 | +1.1 | +2.5 | +4.8 |
| Jun/Jul/Aug | +0.6 | +1.4 | +2.6 | +1.4 | +2.7 | +4.7 | +1.9 | +3.8 | +7 |
| Sep/Oct/Nov | +0.5 | +1.1 | +1.8 | +1.1 | +2.1 | +3.5 | +1.7 | +3 | +5.2 |





Table 2a: Estimated total volume of precipitation (P), evapotranspiration (ET), recharge (R) and discharge (Q) under varied climate conditions (median climate scenarios of 2030, 2050 and 2070 as well as the uncertainty of the climate scenario of 2070) for the simulated time period of 330 days and their units are MCM.

| climate condition | P | ET | R | | | Q | | |
|---|---|---|---|---|---|---|---|---|
| | catchment | catchment | catchment | catchment | QS | QA | QE | SR |
| current | 52.79 | 2.39 | 44.02 | 49.41 | 10.09 | 21.81 | 11.29 | 6.23 |
| 2030 | 50.58 | 2.52 | 42.08 | 47.32 | 9.88 | 21.35 | 10.26 | 5.83 |
| 2050 | 48.48 | 2.66 | 40.15 | 45.33 | 9.69 | 20.99 | 9.14 | 5.51 |
| 2070 | 46.97 | 2.77 | 38.76 | 43.91 | 9.56 | 20.89 | 8.17 | 5.28 |
| 2070 max | 53.15 | 3.34 | 43.74 | 49.33 | 10.15 | 23.96 | 10.09 | 6.04 |
| 2070 min | 38.87 | 2.35 | 32.10 | 36.80 | 8.80 | 17.70 | 5.27 | 4.28 |

Table 2b: Estimated temporary water storage volumes (S) for the whole catchment, karst area, non-karst area and karst aquifer at time step of 2665 (March) and 7896 (October) under varied climate conditions (median climate scenarios of 2030, 2050 and 2070 as well as the uncertainty of the climate scenario of 2070) and their units are MCM.

| climate condition | S | | | | | | | |
|---|---|---|---|---|---|---|---|---|
| | at time step of 2665 (March) | | | | at time step of 7896 (October) | | | |
| | whole catchment | karst area | non-karst area | karst aquifer | whole catchment | karst area | non-karst area | karst aquifer |
| current | 6.20 | 3.99 | 2.21 | -1.77 | 0.99 | 0.84 | 0.16 | 0.84 |
| 2030 | 5.97 | 3.82 | 2.15 | -1.70 | 0.73 | 0.58 | 0.15 | 0.58 |
| 2050 | 5.37 | 3.41 | 1.96 | -1.58 | 0.49 | 0.34 | 0.15 | 0.34 |
| 2070 | 4.23 | 2.54 | 1.69 | -1.38 | 0.29 | 0.14 | 0.15 | 0.14 |
| 2070 max | 5.28 | 3.32 | 1.97 | -0.41 | 0.67 | 0.52 | 0.15 | 0.52 |
| 2070 min | 0.19 | -0.10 | 0.28 | -1.68 | -0.29 | -0.43 | 0.14 | -0.43 |




Figure 1



Figure 1 a) Location of the study area, b) digital elevation model with grid size 100 m × 100 m for the studied catchment and its surrounding area with weather stations used for the interpolation of meteorological parameters and c) model configuration (modified after Chen and Goldscheider 2014).


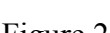


Figure 2

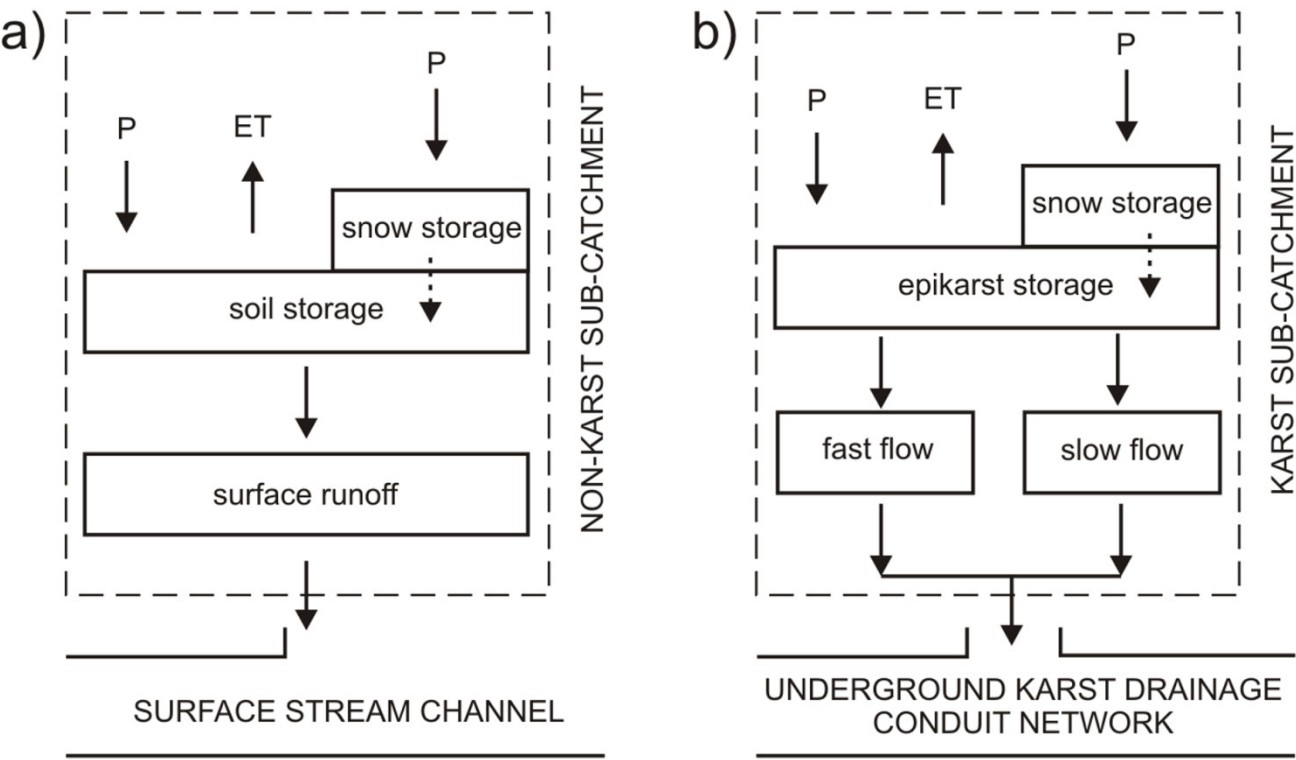

Figure 2 a) model concept for the sub-catchments in the non-karst area and b) model concept for the sub-catchments in the karst area.





Figure 3

| | Hydrodynamic conditions | Simulation period | Target model domain | Observation time series |
|---|---|---|---|---|
| Step 1 | MF - HF | No melt | Unsaturated zone + drainage network | QS, QA, QE and SR |
| Step 2 | MF - HF | Snow and melt | Snow storage | QS and QA |
| Step 3 | LF | Snow and melt | Snow storage + unsaturated zone + drainage network | QS and QA |
| Step 4 | LF - HF | Snow and melt / No melt | Snow storage + unsaturated zone + drainage network | QS, QA, QE and SR |

Constraining parameter ranges

Figure 3 Strategy for the multi-step model calibration, where LF / MF / HF are for low / medium / high flow conditions.




Figure 4

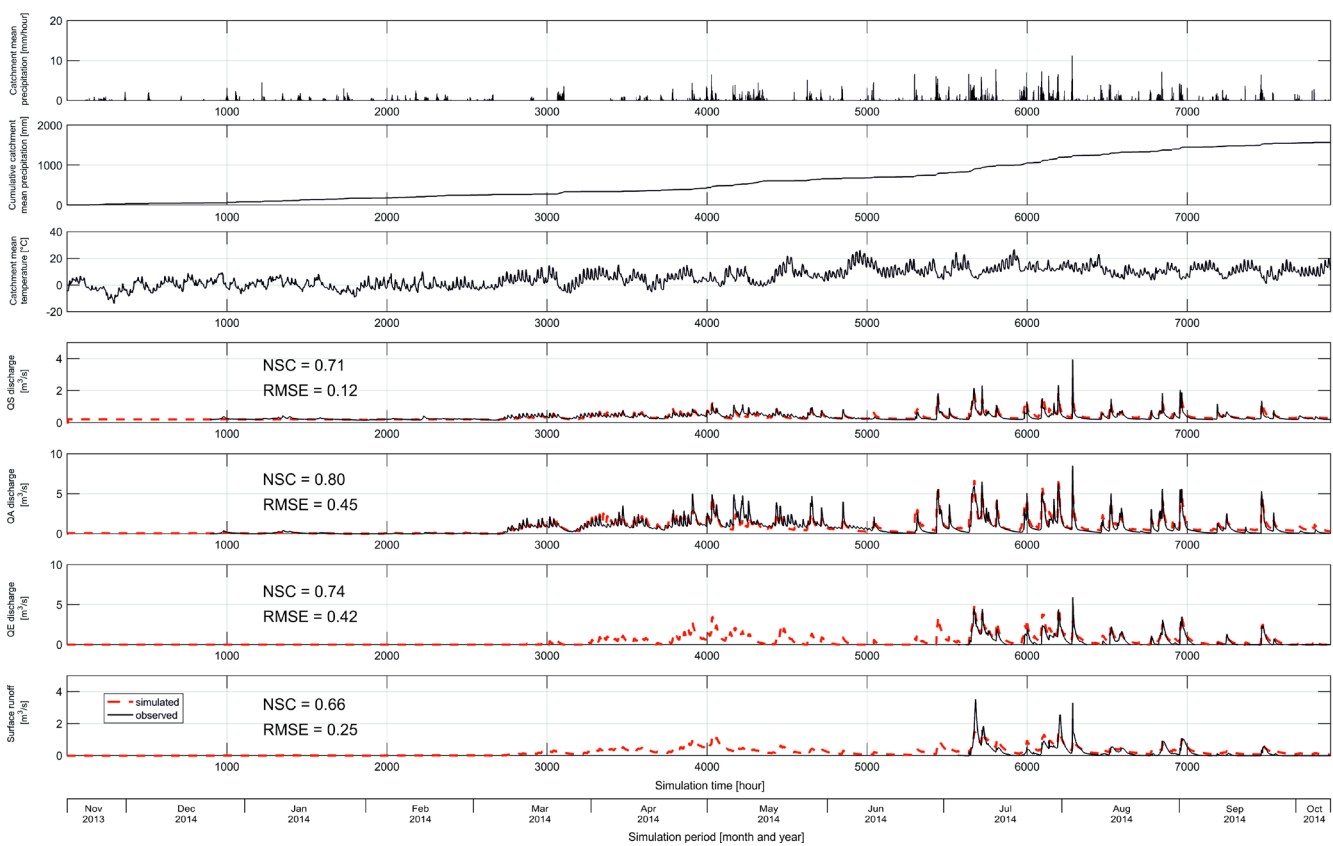

Figure 4 Observed and simulated discharge of four spatially-distributed model outlets QS, QA, QE and SR using the best calibrated model parameter set for the period November 2013 – October 2014. Additionally, the mean catchment precipitation and temperature for the same period are shown.




Figure 5

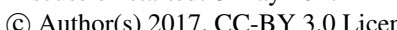

Figure 5 Estimated cumulative volumes of precipitation, evapotranspiration, recharge and discharge for the studied catchment for the period November 2013 – October 2014 on an hourly time step in million cubic meters (MCM).





Figure 6

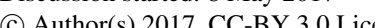

Figure 6 Estimated temporary water storage volumes for the whole catchment, karst area, non-karst area and karst aquifer for

the period November 2013 – October 2014 on an hourly time step in million cubic meters (MCM).



Figure 7

Figure 7 Impacts of the median climate scenarios (q0.5) for 2030, 2050 and 2070 as well as the uncertain climate scenarios (1000 random sampled combinations) for 2070 on the simulated precipitation, evapotranspiration, recharge and discharge for the studied catchment.


Figure 8

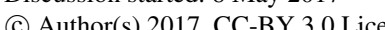


Figure 8 Impacts of the median climate scenarios (q0.5) for 2030, 2050 and 2070 as well as the uncertain climate scenarios (1000 random sampled combinations) for 2070 on the simulated water storages of the whole catchment, karst area, non-karst area and karst aquifer.


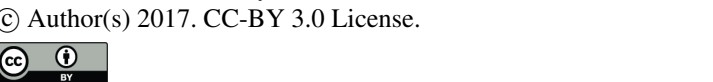

Figure 9

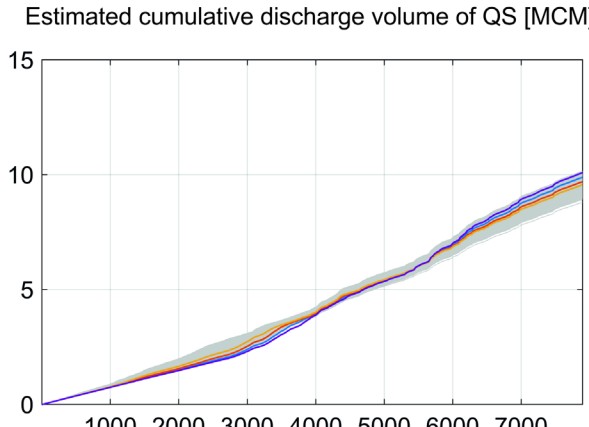
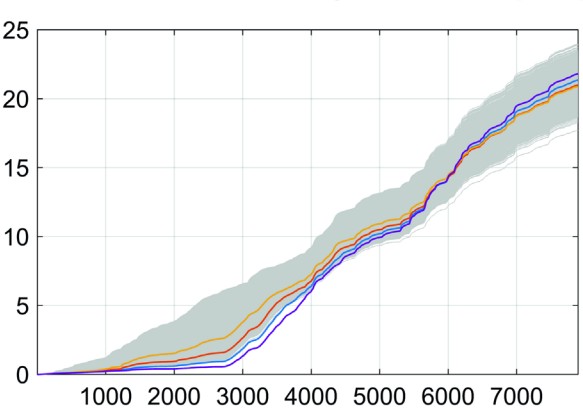
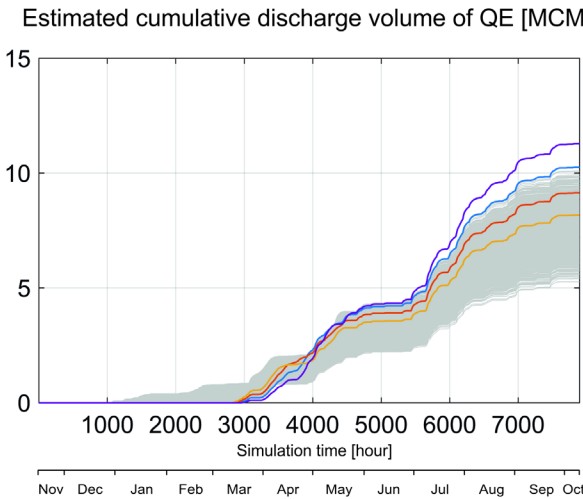
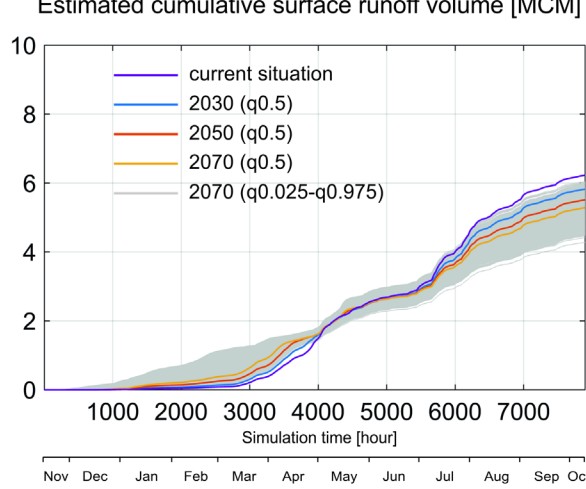

Figure 9 Impacts of the median climate scenarios (q0.5) for 2030, 2050 and 2070 as well as the uncertain climate scenarios (1000 random sampled combinations) for 2070 on the simulated discharge of QS, QA, QE and surface runoff from the non-karst area.