# Peer review of "Dynamics of water fluxes and storages in an Alpine karst catchment under current and potential future climate conditions"

_Hydrology and Earth System Sciences, 2017_

## Referee Comment (RC1) · Anonymous Referee #1 · 20 Jun 2017

The subject paper presents a model based on a previous model (Chen et al., 2014) for the northern Alps on the Germany-Austria border, adding new developments as the presence of the non-karst area and the surface runoff, the slow flow and the snow accumulation and melting The work seems fine for me but, however, the manuscript is hard to read itself and it looks very focused on the study area, that is why I must say that the manuscript is not suitable for publication unless it was revised in some aspects.

The model is based on the one developed by Chen et al., 2014 but, are there more differences between both models apart of the new developments? The basic setting of the model should be explained in the manuscript: what boundary conditions do

the authors use and why? What are the equations that are solved... I miss also a brief discussion about the differences between both models in terms of hydrodynamics (how the new developments change the results and why they should be included). In fact, it looks like the new model fits the discharges considerably worse than in the previous one.

The authors also present some climate scenarios in order to evaluate the behavior of the system and the connections between climate change and subsurface dynamics.in karst aquifers. This could be a research of broad interest however, again, the discussion is hard to follow without going through literature and the conclusions are very focused on the study area. May be it would be interesting to compare this work with some other studies of karst aquifers dynamics.

Talking about the "extremely dry conditions" in 2070, it is not clear to me how the authors introduce the baseflow, is it a constant value as in the work of Chen et al., 2014 or is it different now? Please, explain.

Regarding the conclusions, the authors claim that "the results demonstrate that the spatiotemporal distribution of water fluxes and storages is controlled by surface hydrological setting" but also that "the results should only be applied to understand the relationship between the hydrological processes within the studied catchment and potential climate change patterns". The conclusions are valid but they do not seem to be relevant for the general hydrogeological knowledge. The authors should try to generalize them. It would be interesting to evaluate how a karst system could be affected by climate change, why it is different the affection of climate change over a karst aquifer, are karst aquifers more vulnerable than no karstified ones?
* * *

---

## Referee Comment (RC2) · Anonymous Referee #2 · 30 Jul 2017

Chen et al., Dynamics of water fluxes and storages in an Alpine karst catchment under current and potential future climate conditions, HESS-2017-216

Chen et al. simulate the water storages in a karst catchment using a distributed numerical model. The authors also predicted the hydrology changes under climate changes and stated the significant impacts on karst hydrogeological responses. Overall, this paper is novel and well written, so I would recommend HESS publish after a major revision.

Here are some of my comments:

P1L12: I suggest the authors to provide a brief introduction of distributed numerical

model when mentioned this term, since the readers might need some help to understand this word. If the authors do not want to have a description in the abstract, simply talked about the details of distributed numerical model later.

P1L19-27: I would expect a few sentences to specifically highlight that why the study of karst catchment is important, and the water resources in karst region is vulnerable under future climate change conditions. What is the difference of hydrological responses between karst and non-karst catchment? What it the scientific merit in this study?

P2L10-11: I would say the lack of input variables and model parameters in hydrology model is not only a challenge in Alpine, but also for the hydrological models in other regions. And, is "spatially-distributed model" equal to "distributed numerical model"? Just try to keep consistent and avoid misunderstanding.

P3L1-3: I doubt if it is appropriate to say the relationship between subsurface hydrology and climate has not been considered in detailed. Numerous papers have tried to addressed the relationship, if you simply google some keywords. I would recommend the authors take a look at the review paper by Taylor et al., 2013, Groundwater water and climate change, Nature Climate Change, DOI: 10.1038/NCLIMATE1744

P4L29: What is the source of meteorological data did the authors use? What parameters does the model need? It seems that the authors use the in-situ observational data from the meteorology stations. Since the authors mentioned the uncertainty issues of weather forcing at the very end of this paper, I suggest the authors take a look at the climatological/meteorological reanalysis dataset. I'm not familiar with the reanalysis product in Europe, but I'm sure there are some datasets (eg. ERA-Interim) or global datasets you can use.

P5Ln4-12: The authors mentioned that the melt factor and radiation coefficients were estimated by model calibration. What observational data did you use to calibrate the parameters? And also, I'm afraid the snow accumulation and melting equations are too simple, especially considering the importance of snow melt in this study. Could you

validate the accuracy of snow accumulation and melting?

P5Ln20: I'm not sure if the calibration strategy is an important part in this paper. I would recommend the authors address the physics of the distributed numerical model rather than the calibration.

P6Ln25-26: How did you include the infiltration in the storage calculation for the non-karst area? Please explain or consider to rewrite this sentence.

P7Ln20: It seems the authors use one year (water year 2014) simulation as the base to make future projection with the changing precipitation and ET forcing. Should you consider run the distributed numerical model in multiple years? The climatological average hydrological responses from the model should be used here, if long-term data are available.

P7Ln23: The total volume of mass water does not make sense to the readers who are not familiar to the study area. Is it better to use flux unit (m, or m/day, divided by the area of study domain) to represent the water mass? (I would say it's an open question for the the authors to think about). And also, I highly recommend the authors plot the mass budget of each component instead of using the time-series plot in Figure 5.

P8Ln14: The references of projected precipitation and ET are missing? Are these predictions estimated from an earth system model?

P8Ln20: How did you estimate the snowmelt and snow storage? Did you simply compute from the snowmelt equations in Sect. 3.3 or from an earth system model? Please explain and provide more information.

P9Ln19-20: The "spatial-temporal" distribution is one of the major finding and novel point in this study. I recommend the authors address this point more.

P10Ln17-18: What is the statistics of surface runoff responses to heavy rainfall events? Overestimation or underestimation? How did you compare? In general, I don't think you can directly compare the simulated surface runoff with streamflow measurement.

P10Ln27: Should be "hydrological process sensitivities"?

P11Ln7-9: The different hydrological responses at karst springs are interesting in this study. I recommend the authors highlight the importances of elevation dependency and the permeability of aquifer in water storage capacity and streamflow discharge.

Additional comments: Looking at the reviewer #1 comment, I argee that a more detailed description of the distributed numerical model should be included, and the difference between this model and the previous paper should be highlighted.

———————————————

---

## Short Comment (SC1) · 19 Sep 2017

The subject paper discusses an advance in the ability to model an integrated model of an Alpine watershed that was developed to assess potential impacts to the dynamics of the watershed due to climate change. The study area is an Alpine watershed and, as such, exhibits particular features and dynamics endemic to watersheds greatly affected by orographic dynamics and a complex recharge system subject to a snowpack and snowmelt dynamic. Being an Alpine climate, water is seasonally held in storage as snow.

Of concern is how climate change will alter the durations of when water is held in storage as snow or in the subsurface in the liquid phase. Future climate changes will have two impacts on recharge: (i) the sheer quantity of precipitation and (ii) change of precipitation from snow to rain due to increases in temperature. Increased temperatures will also result in increased evapotranspiration. Model simulations indicate that total recharge (and discharge) decreases under all evaluated future climate conditions.

The study area is only 35 km2, but varies in elevation from 1,000 m to 2,230 m. Mean precipitation at 1,240 m is 1,836 mm/yr. The study domain has two sub-areas, one is a karst area with a subsurface drainage system. The other is a non-karst area with a surface drainage system. The conceptual and numerical model has a number of sub-basins. The karst sub-basins incorporate a conduit/diffuse flow regime. Discharge is measured hourly at four springs, at varying elevations: 1,035 m, 1,080 m, 1,120 m, and 1,122 m, but only for 11/2013-10/2014 at the two lower springs and 7-10/2014 for the two higher springs. The duration for which data are available is not long. This may be the source for the excessively high estimation of recharge percentage of precipitation.

Air temperature, precipitation, and relative humidity were measured at nine stations across the study domain. The authors cite Wending and Muller (1984) as the source for the Turc-Ivanov approach to calculate evapotranspiration. This may be an original source for this approach, however it is not readily available (and not in English). The authors might consider adding Conradt et al. (2013) (HESS) as an additional more accessible citation on this. Precipitation, temperature, and relative humidity are measured at nine weather stations. Each data type is input at a 100 m x 100 m grid using combined inverse distance weighting and linear regression gridding. I suspect this is key to the ability of the authors to match discharge at the four gauging stations as illustrated in Figure 4.

Simulations considered incremental decreases in precipitation in conjunction with incremental increases in evapotranspiration. These changes in input resulted in decreases to recharge. Frie (2004) provides projections for temperature increases by 2030, 2050, and 2070. Gobiet et al. (2014) is cited as a possible source for climate

projections.

The authors acknowledge that their calculation that recharge is 95.5% of precipitation may be overestimated. They noted that Malard et al. (2016) estimate average infiltration rates for mountainous karst catchments across Switzerland vary between 60% and 90%. The source of the over-estimation may be inherent in the pipe network model used to replicate groundwater flow coupled with the optimization routine used to estimate recharge values. The pipe network software package does not allow for matrix-pipe hydraulic communication. Adding the pipe network to the analysis is an advancement to Alpine water-resource assessment, but not including matrix-pipe communication is a limitation. This could be addressed in future work.

Given the high density of precipitation measurement stations (nine) in such a small area, I would think that that precipitation is fairly well constrained. Likewise, all discharge from the basin is measured at the four springs. Unless the basin water budget is not consistent with this conceptualization, it should be possible to provide an independent estimate of recharge using this simplified water budget analysis.

Authors should define FDC. I believe it is Flow Duration Curve, but not positive.

Are tables 1a and 1b from Frei (2004)? If so, please provide citation. Details on the distributed karst catchment model used in this study are in Chen and Goldscheider (2014). The model was derived from a distributed hydrologic-hydraulic water quality simulation model - Storm Water Management Model (SWMM versions 5.0). The GW system was modeled as a pipe network with no hydraulic communication between the matrix and the conduits. Recharge was input as focused point sources. This modeling approach is possible, in part, due to the relatively small size of the study domain (35 km2) and by virtue of the fact that the pipe network (i.e., conduits) has been well defined using tracer tests (Gremaud and Goldscheider, 2010). This limits the ability of the model to be used for predictive simulations. It would be interesting, that given the fact that the conceptual model of the system is fairly well known, if an alternative

mechanistic GW flow model could be developed to test the predictive ability of a model.

The authors used 5000 Latin hypercube runs to determine best fit input parameters. Flow predictions at the four gauges were quite good. There is the risk the authors over-parameterized the model domain. Given the modest duration of data, there was no opportunity to validate the model for time series data not used in the calibration.

Two recommendations for future work on this watershed. (i) Validate the model using future data series. (ii) Develop a mechanistic model to replicate GW flow. This should allow for independent confirmation of the conceptual model and state variable properties currently estimated.
* * *

---

## Author Comment (AC1) · 30 Oct 2017

Reply to comments by an anonymous referee (Reviewer 1) on the manuscript "Dynamics of water fluxes and storages in an Alpine karst catchment under current and potential future climate conditions" by Zhao Chen et al.

Summary of the reviewer: The subject paper presents a model based on a previous model (Chen et al., 2014) for the northern Alps on the Germany-Austria border, adding new developments as the presence of the non-karst area and the surface runoff, the slow flow and the snow accumulation and melting. The work seems fine for me but, however, the manuscript is hard to read itself and it looks very focused on the study

area, that is why I must say that the manuscript is not suitable for publication unless it was revised in some aspects.

Reply: We thank the reviewer for her/his useful and valuable comments that will help to improve the manuscript, in particular its readability. We will carefully rework the entire text in order to improve its structure and intelligibility, also with the help of a native speaker. All suggestions will be taken into account. According to her/his comments, we will perform the following changes.

Comments:

The model is based on the one developed by Chen et al., 2014 but, are there more differences between both models apart of the new developments? The basic setting of the model should be explained in the manuscript: what boundary conditions do the authors use and why? What are the equations that are solved? I miss also a brief discussion about the differences between both models in terms of hydrodynamics (how the new developments change the results and why they should be included). In fact, it looks like the new model fits the discharges considerably worse than in the previous one.

Reply: We will add a brief discussion about the differences between the model developed by Chen & Goldscheider 2014 and the model used in this study. Also, we will include a more detailed description about of the model in this revised manuscript regarding its basic setting, boundary conditions and the underlying equations.

The authors also present some climate scenarios in order to evaluate the behavior of the system and the connections between climate change and subsurface dynamics in karst aquifers. This could be a research of broad interest however, again, the discussion is hard to follow without going through literature and the conclusions are very focused on the study area. May be it would be interesting to compare this work with some other studies of karst aquifers dynamics.

Reply: We thank the reviewer for this valuable suggestion. We will go through the literature and extent the comparison of our results with other karst studies to generalize the discussion and conclusions.

Talking about the "extremely dry conditions" in 2070, it is not clear to me how the authors introduce the base flow, is it a constant value as in the work of Chen et al., 2014 or is it different now? Please, explain.

Reply: We thank the reviewer for pointing this out. This is different to the work by Chen & Goldscheider 2014. We used the linear reservoir approach by Hartmann et al 2011 to simulate base flow (also mentioned as slow flow in this manuscript), which is depending on the recharge process and influenced by the chanced climate conditions. We will explain this by comparing the differences between the model developed by Chen & Goldscheider 2014 and the model used in this study (see our answer above: the existing model and the new developments will be more clearly explained).

Regarding the conclusions, the authors claim that "the results demonstrate that the spatiotemporal distribution of water fluxes and storages is controlled by surface hydrological setting" but also that "the results should only be applied to understand the relationship between the hydrological processes within the studied catchment and potential climate change patterns". The conclusions are valid but they do not seem to be relevant for the general hydrogeological knowledge. The authors should try to generalize them. It would be interesting to evaluate how a karst system could be affected by climate change, why it is different the affection of climate change over a karst aquifer, are karst aquifers more vulnerable than no karstified ones?

Reply: We thank the reviewer for this valuable suggestion. We will expand the literature review and compare the climate change impact on karst aquifer system with other aquifer types in order to generalize the conclusions and increase the relevance of this study for the general hydrogeological knowledge. In fact, it is true that every karst aquifer system has its unique setting and behavior; therefore, detailed projections cannot be transferred to other karst catchments. However, our general modeling approach and some general conclusions can be transferred to other regions. We will try to show this more clearly and carefully in the revised version of our manuscript, e.g. by differentiating between "local" and "general" conclusions.

References

Chen, Z., Goldscheider, N., 2014. Modeling spatially and temporally varied hydraulic behavior of a folded karst system with dominant conduit drainage at catchment scale, Hochifen–Gottesacker, Alps. J. Hydrol. 514, 41–52. 10.1016/j.jhydrol.2014.04.005.

Hartmann, A., Kralik, M., Humer, F., Lange, J., Weiler, M., 2011. Identification of a karst system's intrinsic hydrodynamic parameters: upscaling from single springs to the whole aquifer. Environ. Earth Sci. 65, 2377–2389. 10.1007/s12665-011-1033-9.

---

## Author Comment (AC2) · 30 Oct 2017

Reply to comments of an anonymous referee (Reviewer 2) on the manuscript "Dynamics of water fluxes and storages in an Alpine karst catchment under current and potential future climate conditions" by Zhao Chen et al.

Summary of the reviewer: Chen et al. simulate the water storages in a karst catchment using a distributed numerical model. The authors also predicted the hydrology changes under climate changes and stated the significant impacts on karst hydrogeological responses. Overall, this paper is novel and well written, so I would recommend HESS publish after a major revision.

Reply: We thank the reviewer for her/his positive, very useful and constructive comments that will contribute to improve the manuscript. Most of the referees' remarks will be taken into account. According to her/his comments, we will perform the following changes.

Comments:

P1L12: I suggest the authors to provide a brief introduction of distributed numerical model when mentioned this term, since the readers might need some help to understand this word. If the authors do not want to have a description in the abstract, simply talk about the details of distributed numerical model later.

Reply: We will give more details about the term "distributed numerical model" in the introduction.

P1L19-27: I would expect a few sentences to specifically highlight that why the study of karst catchment is important, and the water resources in karst region are vulnerable under future climate change conditions. What is the difference of hydrological responses between karst and non-karst catchment? What it the scientific merit in this study?

Reply: We will highlight the importance of karst catchment study and mention that karst water resources are especially vulnerable under changing climate conditions. Also we will explain shortly the difference of hydrological responses between karst and non-karst catchment in order to elaborate the significance of this study.

P2L10-11: I would say the lack of input variables and model parameters in hydrology model is not only a challenge in Alpine, but also for the hydrological models in other regions. And, is "spatially-distributed model" equal to "distributed numerical model"? Just try to keep consistent and avoid misunderstanding.

Reply: The reviewer's remarks are reasonable. We will modify the text accordingly.

P3L1-3: I doubt if it is appropriate to say the relationship between subsurface hydrol-

ogy and climate has not been considered in detailed. Numerous papers have tried to addressed the relationship, if you simply google some keywords. I would recommend the authors take a look at the review paper by Taylor et al., 2013, Groundwater water and climate change, Nature Climate Change, DOI: 10.1038/NCLIMATE1744.

Reply: Of course we know the important and highly-cited paper by Richard Taylor et al., and we will discuss and cite this paper and other relevant papers in the revised version of our manuscript.

P4L29: What is the source of meteorological data did the authors use? What parameters does the model need? It seems that the authors use the in-situ observational data from the meteorology stations. Since the authors mentioned the uncertainty issues of weather forcing at the very end of this paper, I suggest the authors take a look at the climatological/meteorological reanalysis dataset. I'm not familiar with the reanalysis product in Europe, but I'm sure there are some datasets (e.g. ERA-Interim) or global datasets you can use.

Reply: The suggested ERA-Interim dataset has unfortunately too coarse spatial resolution (80 km) for this study area, which only 35 km2 but varies in elevation from 1000 m to 2230 m. The model needs three input variables, i.e. precipitation, temperature and relative humidity at hourly time steps. These variables were measured at nine weather stations across the study area (Figure 1b). Each meteorological data type is interpolated at a 100 m $\times$ 100 m grid using combined inverse distance weighting and linear regression gridding, in order to consider its spatial variability in the study domain.

P5Ln4-12: The authors mentioned that the melt factor and radiation coefficients were estimated by model calibration. What observational data did you use to calibrate the parameters? And also, I'm afraid the snow accumulation and melting equations are too simple, especially considering the importance of snow melt in this study. Could you validate the accuracy of snow accumulation and melting?

Reply: The study area is high-alpine and difficult to access. The observation data is

strongly limited and we do not have direct snow observation to validate the accuracy of simulated snow accumulation and melting. Due to the lack of data, we applied the HBV snow routine and modified the snow melt equation after Hock (1999). His study demonstrated that the modified snow melt equation is able to simulate realistic hourly melt and its spatial pattern in complex topography. In order to achieve an effective calibration of the snow routine used in this study, we have developed the multi-step model calibration procedure described in section 3.4.2. The discharge time series from November 2013 to June 2014 (whereas snow accumulation and melting dominated) measured by the gauging stations at QS and QA were used for the calibration. However, the simulated snow melt and especially the accumulation are associated with uncertainty. We will consider this critical point and discuss the consequences of choosing a conceptual-type snow simulation approach.

P5Ln20: I'm not sure if the calibration strategy is an important part in this paper. I would recommend the authors address the physics of the distributed numerical model rather than the calibration.

Reply: We thank the reviewer for this recommendation. We will include a more detailed description about the model used in this study (see response letter to reviewer 1) and address in this part the physics represented in the model. We do think that the proposed calibration strategy is an import part for this study though, which helped us to achieve an effective model calibration.

P6Ln25-26: How did you include the infiltration in the storage calculation for the non-karst area? Please explain or consider rewriting this sentence.

Reply: The surface runoff from the non-karst area can infiltrate into the underground karst drainage network due to the conduits C34 – C38 constructed in the upper part of the valley (Figure 1c). The flow through the conduits into the underground karst drainage network is considered as the infiltration (allogenic recharge) into the karst aquifer. We could quantify explicitly the infiltration and use Equation 7 to calculate the

storage for the non-karst area.

P7Ln20: It seems the authors use one year (water year 2014) simulation as the base to make future projection with the changing precipitation and ET forcing. Should you consider run the distributed numerical model in multiple years? The climatological average hydrological responses from the model should be used here, if long-term data are available.

Reply: We agree with the reviewer that the distributed numerical model should be run in multiple years to make projections with the changing precipitation and ET forcing. However the study area is high-alpine and difficult to assess. The complete hydrological monitoring system has been operating since 2012 and we were lucky to obtain at least one complete hydrological year data, which can be used for this modeling study. To better assess the model, despite of one hydrological year data, we proposed performing a split sample test by using multiple bootstrapping of subsets of the observation period (section 3.2). This approach is already used by Hartmann et al (2012). We will include and discuss the test result in the revised manuscript.

P7Ln23: The total volume of mass water does not make sense to the readers who are not familiar to the study area. Is it better to use flux unit (m, or m/day, divided by the area of study domain) to represent the water mass? (I would say it's an open question for the authors to think about). And also, I highly recommend the authors plot the mass budget of each component instead of using the time-series plot in Figure 5.

Reply: The reason why we used the total volume of mass water is the flow calculation referred to different areas (karst area and / or non-karst area). To better compare the flow components with each other, we decided to use the total volume of mass water as unit. Additionally, we will adjust Figure 5 for increased simplicity (e.g. plot the mass budget of each component).

P8Ln14: The references of projected precipitation and ET are missing? Are these predictions estimated from an earth system model?

Reply: The "current situation" is the model simulation based on the existing hydrological data from 2013 to 2014. We will clarify this in the revised version of the manuscript.

P8Ln20: How did you estimate the snowmelt and snow storage? Did you simply compute from the snowmelt equations in Sect. 3.3 or from an earth system model? Please explain and provide more information.

Reply: The description about the maximal snow storage is referred to the simulation time step 3109 (Figure 8). There we estimated the snowmelt using the equations described in section 3.3 and assuming that the positive water storage value is equal the snow storage, which is admittedly a quite pragmatic approach. In fact, the maximum snow storage is clearly underestimated at that simulation time step, as we did not consider the temporary water storage volume in the karst aquifer, which is negative. In the revised manuscript, we will show the calculated snow storage obtained from the HBV snow routine, which is already implemented in the current model. Accordingly, we will modify the text and add a new figure regarding the snow storage. Also, we will add some more discussion about the uncertainties that resulted from our approach.

P9Ln19-20: The "spatial-temporal" distribution is one of the major finding and novel point in this study. I recommend the authors address this point more.

Reply: We thank the reviewer for this suggestion and we will follow that recommendation.

P10Ln17-18: What is the statistics of surface runoff responses to heavy rainfall events? Overestimation or underestimation? How did you compare? In general, I don't think you can directly compare the simulated surface runoff with streamflow measurement.

Reply: Here we meant the surface runoff generated from the non-karst area. The comparison between the simulated and observed surface runoff can be done using Figure 4. The model underestimates surface runoff if fluxes are greater than about 2m3/s. We will rewrite this part and avoid the misunderstanding.

P10Ln27: Should be "hydrological process sensitivities"?

Reply: We thank the reviewer for this suggestion. This sounds very elegant. We will use this term.

P11Ln7-9: The different hydrological responses at karst springs are interesting in this study. I recommend the authors highlight the importance of elevation dependency and the permeability of aquifer in water storage capacity and streamflow discharge.

Reply: We thank the reviewer for this recommendation and we will do that.

Additional comments: Looking at the reviewer #1 comment, I agree that a more detailed description of the distributed numerical model should be included, and the difference between this model and the previous paper should be highlighted.

Reply: We will include a more detailed description about the model in this manuscript. Also we will add a brief discussion about the differences between the initial model developed by Chen & Goldscheider 2014 and the extended model used in this study.

References

Chen, Z., Goldscheider, N., 2014. Modeling spatially and temporally varied hydraulic behavior of a folded karst system with dominant conduit drainage at catchment scale, Hochifen–Gottesacker, Alps. J. Hydrol. 514, 41–52. 10.1016/j.jhydrol.2014.04.005.

Hartmann, A., Lange, J., Weiler, M., Arbel, Y., Greenbaum, N., 2012. A new approach to model the spatial and temporal variability of recharge to karst aquifers. Hydrol. Earth Syst. Sci. 16, 2219–2231. 10.5194/hess-16-2219-2012.

Hock, R., 1999. A distributed temperature-index ice- and snowmelt model including potential direct solar radiation. J. Glaciol. 45 (149), 101–111. 10.1017/S0022143000003087.

---

## Author Comment (AC3) · 30 Oct 2017

Reply to comments by Ronald Green on our manuscript "Dynamics of water fluxes and storages in an Alpine karst catchment under current and potential future climate conditions" by Zhao Chen et al.

Reply: We thank Ronald Green for his effort and useful comments that will contribute to further improve the manuscript. According to his comments, we will perform the following changes.

Comments:

[Figure]

The study area is only 35 km2, but varies in elevation from 1,000 m to 2,230 m. Mean precipitation at 1,240 m is 1,836 mm/yr. The study domain has two sub-areas, one is a karst area with a subsurface drainage system. The other is a non-karst area with a surface drainage system. The conceptual and numerical model has a number of sub-basins. The karst sub-basins incorporate a conduit/diffuse flow regime. Discharge is measured hourly at four springs, at varying elevations: 1,035 m, 1,080 m, 1,120 m, and 1,122 m, but only for 11/2013-10/2014 at the two lower springs and 7-10/2014 for the two higher springs. The duration for which data are available is not long. This may be the source for the excessively high estimation of recharge percentage of precipitation.

Reply: We thank Ronald Green for pointing this out. The relatively short time series of discharge measurements is related to the remoteness of this alpine terrain and the difficult accessibility of the springs, which are partly located in steep gorges. We will discuss the possible overestimation of recharge percentage of precipitation below.

Air temperature, precipitation, and relative humidity were measured at nine stations across the study domain. The authors cite Wending and Muller (1984) as the source for the Turc-Ivanov approach to calculate evapotranspiration. This may be an original source for this approach, however it is not readily available (and not in English). The authors might consider adding Conradt et al. (2013) (HESS) as an additional more accessible citation on this.

Reply: We thank Ronald Green for this useful suggestion. We will add Conradt et al. (2013) as the reference for the Turc-Ivanov approach.

Precipitation, temperature, and relative humidity are measured at nine weather stations. Each data type is input at a 100 m x 100 m grid using combined inverse distance weighting and linear regression gridding. I suspect this is key to the ability of the authors to match discharge at the four gauging stations as illustrated in Figure 4.

Reply: We appreciate this observation.

[Figure]

Simulations considered incremental decreases in precipitation in conjunction with incremental increases in evapotranspiration. These changes in input resulted in decreases to recharge. Frei (2004) provides projections for temperature increases by 2030, 2050, and 2070. Gobiet et al. (2014) is cited as a possible source for climate projections.

Reply: For the climate change simulation, only the scenarios by Frei (2014) are considered.

The authors acknowledge that their calculation that recharge is 95.5% of precipitation may be overestimated. They noted that Malard et al. (2016) estimate average infiltration rates for mountainous karst catchments across Switzerland vary between 60% and 90%. The source of the over-estimation may be inherent in the pipe network model used to replicate groundwater flow coupled with the optimization routine used to estimate recharge values. The pipe network software package does not allow for matrix-pipe hydraulic communication. Adding the pipe network to the analysis is an advancement to Alpine water-resource assessment, but not including matrix-pipe communication is a limitation. This could be addressed in future work. Given the high density of precipitation measurement stations (nine) in such a small area, I would think that that precipitation is fairly well constrained. Likewise, all discharge from the basin is measured at the four springs. Unless the basin water budget is not consistent with this conceptualization, it should be possible to provide an independent estimate of recharge using this simplified water budget analysis.

Reply: Our model does consider baseflow resulting from prolonged water storage and slow flow in the rock matrix. This slow flow component is then introduced into the network of karst conduits. It is true that the model is limited in simulating conduit-matrix interaction, but the general configuration of the karst system suggests that at most places and most of the time, the conduits drain the adjacent matrix, because conduits are mostly located in the troughs of plunging synclines. We think that the main reason for the overestimation of recharge is that evapotranspiration is underestimated and evaporation from snow is not taken into account. The automatic model calibration constrained the model with the overestimated recharge rate. We will discuss this critical point more in detail in the revised manuscript. Independently, we agree with Ronald Green that the conduit-matrix interaction should be considered in future work to better understand aquifer hydraulics and contaminant transport.

Authors should define FDC. I believe it is Flow Duration Curve, but not positive.

Reply: Yes, it is Flow Duration Curve and we will define it in the revised manuscript.

Are tables 1a and 1b from Frei (2004)? If so, please provide citation. Details on the distributed karst catchment model used in this study are in Chen and Goldscheider (2014). The model was derived from a distributed hydrologic-hydraulic water quality simulation model - Storm Water Management Model (SWMM versions 5.0). The GW system was modeled as a pipe network with no hydraulic communication between the matrix and the conduits. Recharge was input as focused point sources. This modeling approach is possible, in part, due to the relatively small size of the study domain (35 km2) and by virtue of the fact that the pipe network (i.e., conduits) has been well defined using tracer tests (Gremaud and Goldscheider, 2010). This limits the ability of the model to be used for predictive simulations. It would be interesting, that given the fact that the conceptual model of the system is fairly well known, if an alternative mechanistic GW flow model could be developed to test the predictive ability of a model.

Reply: Yes, tables 1a and 1b are based on Frei (2004). We will add the reference. The tracer tests were not done by Gremaud & Goldscheider, who worked in another alpine karst system, but by Goldscheider (2005) and Göppert & Goldscheider (2008). Concerning conduit-matrix-interaction, see our reply above: It is true that the model does not simulate bidirectional hydraulic interaction, but it does consider matrix flow by means of a reservoir model that accounts for water storage and slow flow in the adjacent rock matrix. Most of the conduits are located in the troughs of synclines, so the geological structure of this karst system justifies the selected modeling approach. Still, we

agree with Ronald Green that alternative modeling approaches (such as MODFLOW-CFP) could further improve the understanding of this karst system. We are working on that.

The authors used 5000 Latin hypercube runs to determine best fit input parameters. Flow predictions at the four gauges were quite good. There is the risk the authors over-parameterized the model domain. Given the modest duration of data, there was no opportunity to validate the model for time series data not used in the calibration.

Reply: This is a fair comment, and we are aware of this problem. However, in order to minimize the risk of over-parameterization, we have developed a multi-step model calibration strategy (please see section 3.4.2). Furthermore, to better assess the model, despite the relatively short observation period of only one hydrological year, we propose to perform a split sample test by using multiple bootstrapping of subsets of the observation period (section 3.2). This approach is already used by Hartmann et al (2012). We will include and discuss the test result in the revised manuscript.

Two recommendations for future work on this watershed. (i) Validate the model using future data series. (ii) Develop a mechanistic model to replicate GW flow. This should allow for independent confirmation of the conceptual model and state variable properties currently estimated.

Reply: We thank Ronald Green for these two valuable recommendations. We are thinking to simulate the catchment by applying the MODFLOW-CFP, which is able to consider conduit-matrix interaction and can also be used for transport simulations, which is very useful in this case, because we have data from over 16 tracer tests done in this alpine karst system.

References

Conradt, T., Wechsung, F., Bronstert, A., 2013. Three perceptions of the evapotranspiration landscape: comparing spatial patterns from a distributed hydrological model,

remotely sensed surface temperatures, and sub-basin water balances. Hydrol. Earth Syst. Sci. 17, 2947–2966. 10.5194/hess-17-2947-2013.

Frei, C., 2004. Die Klimazukunft der Schweiz - Eine probabilistische Projektion. http://www.occc.ch/Products/CH2050/CH2050-Scenarien.pdf.

Goldscheider, N., 2005. Fold structure and underground drainage pattern in the alpine karst system Hochifen-Gottesacker. Eclogae geol. Helv 98 (1), 1–17. 10.1007/s00015-005-1143-z.

Göppert, N., Goldscheider, N., 2008. Solute and Colloid Transport in Karst Conduits under Low- and High-Flow Conditions. Ground Water 46 (1), 61–68. 10.1111/j.1745-6584.2007.00373.x.

Hartmann, A., Lange, J., Weiler, M., Arbel, Y., Greenbaum, N., 2012. A new approach to model the spatial and temporal variability of recharge to karst aquifers. Hydrol. Earth Syst. Sci. 16, 2219–2231. 10.5194/hess-16-2219-2012.

---

## Author Comment (AC4) · 7 Feb 2018

Reply to Editor' comments regarding the response to the Reviewer 1 on the manuscript "Dynamics of water fluxes and storages in an Alpine karst catchment under current and potential future climate conditions" by Zhao Chen et al.

Editor' comments:

Potential overlap with the paper by Chen and Goldscheider (JHyd, 2014). We are trying to discourage the practice of publishing similar material several times.

Answer: There is no overlap with the paper by Chen & Goldscheider (2014). The cur-

rent study presents entirely new research questions, new data, new results and new conclusions. The model tested and evaluated in this manuscript is a further development of the model introduced in the study by Chen & Goldscheider (2017). The major novel aspects include:

1) The new research questions (presented in the introduction) focus on variable flow and storage over the entire hydrologic year, including the period of snow accumulation and snowmelt, and on potential impacts of climate change on the dynamic water balance. Accordingly, extensive new data series (including winter and spring) were considered. 2) The updated model adopts the HBV-snow routine and is able to simulate snow storage and snowmelt and their influence on groundwater recharge processes. 3) The earlier model considers baseflow / slow flow as a constant value, which is insufficient for long-term climate-change impact predictions. In the updated model, we applied the linear reservoir approach by Hartmann et al. 2011 to simulate transient slow-flow components, depending on groundwater recharge and recession coefficients. 4) The laterally adjacent and hydrogeologically connected non-karst area is included in the current model domain. The updated model is able to simulate variable infiltration of surface runoff from the non-karst area into the underground karst drainage network. 5) In the updated model, the spatial discretization of the catchment area is much finer by using the elevation bands approach, which allows for a better representation of the spatial variability of meteorological variables.

In response to comments by reviewer 1 about the specifics of your model, you reply "We will add a brief discussion".

Answer: Our model is construced by using a hybrid-struture: a combined lumped-parameter and distributed-parameter approach. Basically, the lumped-parameter model represents water storage and drainage in the soil and epikarst. The distributed paramter model represents the underground karst drainage network in the karst area, and the network of surface streams in the non-karst area; these linear structes drain the flow generated from the lumped paramter model. Due to the new developements,

the current model is able to simulate simultaneously all system outlets for a complete hydrological year, including priods of snow acculumuation, snowmelt and rainfall; additionally, the current model is able to reproduce system discharge behavior during drought periods, as the system baseflow was implemented as a function of groundwater recharge and recession coeffient. In this study, the simulation started in late autumn (November 2013), during very low flow conditon. The discharge of QS during this time consists of slow flow components from the karst area. This hydrologic state was used to define the initial model condition.

In response to the comment of also reviewer 1 about the unclear relevance of your findings for the general hydrogeological knowledge, you again reply "we will".

Answer: Reviewer 1 wrote "The conclusions are valid but they do not seem to be relevant for the general hydrogeological knowledge. The authors should try to generalize them".

We are thankful for this comment, and we will include new and more generalized conclusions, which also demonstrate the broader relevance and transferability of our modeling approach and scientific findings. The general conclusions can be summarized as follows:

Because of their unique hydraulic characteristics, karst aquifers respond faster and stronger on hydrological events and seasonal variations, including snow accumulation and melting, than other types of aquifers. The frequency and intensity of extreme events and the seasonal patterns of precipitation and snow regimes are projected to change in a changing climate. Karst systems are especially vulnerable to these changing hydro-meteorological conditions. However, because of their hydrogeological complexity and hydraulic heterogeneity, every karst system has its individual characteristics, and different karst springs respond differently on changing climatic conditions. Therefore, site-specific investigations are required. The holistic modeling approach presented in our study can be adapted to other types of karst systems and can be

used for studying impacts of climate change on alpine karst water resources.

References:

Chen, Z., Goldscheider, N., 2014. Modeling spatially and temporally varied hydraulic behavior of a folded karst system with dominant conduit drainage at catchment scale, Hochifen–Gottesacker, Alps. J. Hydrol. 514, 41–52. 10.1016/j.jhydrol.2014.04.005.

Chen, Z., Hartmann, A., Goldscheider, N., 2017. A new approach to evaluate spatiotemporal dynamics of controlling parameters in distributed environmental models. Environ. Modell. Softw. 87, 1–16. 10.1016/j.envsoft.2016.10.005.

Hartmann, A., Kralik, M., Humer, F., Lange, J., Weiler, M., 2011. Identification of a karst system's intrinsic hydrodynamic parameters: upscaling from single springs to the whole aquifer. Environ. Earth. Sci. 65, 2377–2389. 10.1007/s12665-011-1033-9.

---

## Author Comment (AC5) · 7 Feb 2018

Reply to Editor's comment regarding the response to the Reviewer 2 on the manuscript "Dynamics of water fluxes and storages in an Alpine karst catchment under current and potential future climate conditions" by Zhao Chen et al.

Editor's comment: You have left unanswered the question by reviewer 2 about the scientific merit of this study.

Answer: We believe that the main scientific merit of this study is to better understand the highly variable groundwater dynamic in mountainous karst catchments, which can

be highly vulnerable under future changing climate conditions. Our paper presents the first study to investigate potential impacts of climate change on mountainous karst systems by using a combined lumped and distributed parameter modeling approach with consideration of subsurface karst drainage structures. Additionally, this work presents a novel holistic modeling approach, which can be transferred to similar karst systems for studying the impact of climate change on local karst water resources with consideration of their individual hydrogeological complexity and hydraulic heterogeneity. This novelty will be better explained in the revised paper, both in the introduction and in the conclusions.

---

## Author Comment (AC6) · 7 Feb 2018

Reply to Editor's comment regarding the response to Ronald Green on the manuscript "Dynamics of water fluxes and storages in an Alpine karst catchment under current and potential future climate conditions" by Zhao Chen et al.

Editor's comment: The huge fraction of rainfall that recharges has been politely pointed as unusual. I find it inappropriate to reply that "We will discuss the possible overestimation of recharge percentage of precipitation". You must either argue why such percentage (95%) is right or why your model is wrong. Certainly, constraining the model with an overestimated recharge does not sound appropriate, because calibration will

twist all model parameters to yield numerically consistent, but conceptually erroneous, results.

Answer: We agree that a recharge of 95% appears unusual. However, very high recharge rates in mountainous karst areas, ranging between 60% and 90%, are also reported in the literature (e.g. Malard et al 2016). In alpine regions, low temperatures and high precipitation (P) favor low evapotranspiration (ETP). In our test site, and other alpine karst areas, soil and vegetation are almost entirely missing in the elevated parts, and the limestone is extremely karstified, so that water infiltrates directly into open fractures, as can be seen on the photo (Fig. 1, the lower parts of the area are covered by shallow soil and forest, causing higher ETP and lower recharge).

We are convinced that our conceptual model is appropriate, as it is based on detailed hydrogeological field investigations, including 18 tracer tests (Goldscheider 2005, Göppert & Goldscheider 2008, Sinreich et al. 2002). The overall size of the karst system, the catchment areas of the individual springs and the general configuration of the underground drainage network are exceptionally well known, which is a major advantage of this test site.

However, the quantification of recharge is associated with uncertainties and the value of 95% is probably an overestimation. Possible reasons include:

1) The interpolation of precipitation is uncertain. Most weather stations used for interpolation are located outside the study area, at lower elevations. Uncertainty depends on the density of observation points and the interpolation method (e.g. Ohmer et al 2017). 2) Discharge quantities during very high flow conditions are also uncertain. We measured continuously water stages at all gauging stations, and we performed numerous flow measurements (salt-dilution method) to establish rating curves, which were used to obtain continuous hydrographs for all system outlets. However, most flow measurements were done during low to moderately high flow conditions, and the rating curves had to be extrapolated for very high flows. Therefore, substantial uncertainties

have to be expected for very high flow conditions (e.g. Baldassarre & Montanari 2009, Coxon et al 2015). 3) Another source of uncertainty is that evaporation from snow was not taken into account in the current model. However, some studies suggest that snow evaporation can be significant in some high elevated catchments (e.g. Leydecker & Melack 2000).

In conclusion, we are convinced that our hydrogeological understanding and conceptual model are appropriate (very low uncertainties). The measurement and interpolation of precipitation, ETP, discharge and, as a consequence, recharge are associated with inherent uncertainties, related to the remoteness and complexity of this alpine karst catchment. The resulting recharge of 95% is probably an overestimation, but it is the best estimation that we can obtain, and it is not far above the reported range of recharge values for high alpine karst catchments.

We will discuss these uncertainties in the revised manuscript.

References:

Baldassarre, G., Montanari, A., 2009. Uncertainty in river discharge observations: a quantitative analysis. Hydrol. Earth Syst. Sci. 13, 913–921.

Coxon, G., Freer, J., Westerberg, I., Wagener, T., Woods, R., Smith, P., 2015. A novel framework for discharge uncertainty quantification applied to 500 UK gauging stations. Water Resour. Res., 51, 5531–5546. 10.1002/2014WR016532.

Goldscheider, N., 2005. Fold structure and underground drainage pattern in the alpine karst system Hochifen-Gottesacker. Eclogae geol. Helv 98 (1), 1–17. 10.1007/s00015-005-1143-z.

Göppert, N., Goldscheider, N., 2008. Solute and Colloid Transport in Karst Conduits under Low- and High-Flow Conditions. Ground Water 46 (1), 61–68. 10.1111/j.1745-6584.2007.00373.x.

Leydecker, A., Melack, J., 2000. Estimating evaporation in seasonally snow-covered

catchments in the Sierra Nevada, California. J. Hydrol. 236 (1-2), 121–138. 10.1016/S0022-1694(00)00290-0.

Malard, A., Sinreich, M., Jeannin, P.-Y., 2016. A novel approach for estimating karst groundwater recharge in mountainous regions and its application in Switzerland. Hydrol. Process. 30 (13), 2153–2166. 10.1002/hyp.10765.

Ohmer, M., Liesch, T., Goeppert, N., Goldscheider, N., 2017. On the optimal selection of interpolation methods for groundwater contouring: An example of propagation of uncertainty regarding inter-aquifer exchange. Advances in Water Resources 109, 121–132.

Sinreich, M., Goldscheider, N., Hötzl, H., 2002. Hydrogeologie einer alpinen Bergsturzmasse (Schwarzwassertal, Vorarlberg). Beit. z. Hydrogeologie 53, 5–20.

**Fig. 1.** View on the summit "Hochifen" and karstic limestone plateau "Gottesacker" in the study region.

---

## Editor Decision (ED1)

The final statement of the paper sounds a bit arrogant (and the last statement is somewhat trivial, it is for others to judge):

Overall, our study provides a better understanding of the highly variable groundwater dynamics in mountainous karst catchments, which can be highly vulnerable under future changing climate conditions. Additionally, this work presents a novel holistic modeling approach, which can be transferred to similar karst systems for studying the impact of climate change on local karst water resources with consideration of their individual hydrogeological complexity and hydraulic heterogeneity.

Instead, you may wish to state something like:
Overall, our study highlights the fast dynamics of mountainous karst catchments, which makes them highly vulnerable to future changing climate conditions.

You may also wish to qualify your statements on page 11

Yet, our quantification of recharge is still associated with uncertainties.
Possible reasons include: 1) the interpolation of precipitation is uncertain. Most weather stations used for interpolation are located outside the study area, at lower elevations. Uncertainty depends on the density of observation points and the interpolation method (e.g. Ohmer et al., 2017). Increase of precipitation with elevation should also be taken into account. 2) Discharge quantities during very high flow conditions are also uncertain. Water stages were continuously measured at all gauging stations, and numerous flow measurements (salt-dilution method) were performed to establish rating curves, which were used to obtain continuous hydrographs for all system outlets. However, most flow measurements were done during low to moderately high flow conditions, and the rating curves had to be extrapolated for very high flows. Therefore, substantial uncertainties have to be expected for very high flow conditions (e.g. Baldassarre & Montanari, 2009; Coxon et al., 2015). 3) Another source of uncertainty is that sublimation from snow was not taken into account in the current model. However, some studies suggest that snow evaporation can be significant in some high elevated catchments (e.g. Leydecker & Melack, 2000).

---

## Author Response (AR2)

Response Letter concerning the manuscript:

Dynamics of water fluxes and storages in an Alpine karst catchment under current and potential future climate conditions (hess-2017-216)

Zhao Chen, Andreas Hartmann, Thorsten Wagener, Nico Goldscheider

We thank Editor for his effort! Now, we have considered all his recommendations in our corrected manuscript (highlighted in blue).